# Within-Season Changes in Land-Use Impact Pest Abundance in Smallholder African Cassava Production Systems

**DOI:** 10.3390/insects12030269

**Published:** 2021-03-22

**Authors:** Andrew Kalyebi, Sarina Macfadyen, Andrew Hulthen, Patrick Ocitti, Frances Jacomb, Wee Tek Tay, John Colvin, Paul De Barro

**Affiliations:** 1National Crops Resources Research Institute, P.O. Box 7084, Kampala, Uganda; ocittipatrick@gmail.com; 2Mikocheni Agricultural Institute, Dares Salaam 6226, Tanzania; 3CSIRO, Clunnies Ross Street, Acton 2601, Australia; sarina.macfadyen@csiro.au (S.M.); f.jacomb@gmail.com (F.J.); weetek.tay@csiro.au (W.T.T.); 4CSIRO, Ecosciences Preceinct, Dutton Park QLD, Brisbane 4001, Australia; andrew.hulthen@csiro.au (A.H.); paul.debarro@csiro.au (P.D.B.); 5NRI, University of Greenwich, Chatham, Maritime, Kent ME4 4TB, UK; j.colvin@greenwich.ac.uk

**Keywords:** *Bemisia tabaci*, temporal dynamics, landscape change, pest abundance, ecosystem services

## Abstract

**Simple Summary:**

Cassava is one of the most important subsistence crops for smallholder farmers in East and central Africa. Cassava is increasingly infested by the whitefly *Bemisia tabaci* that transmits two diseases that in combination with the direct feeding damage cause significant yield losses. Land-use changes due to crop rotations can impact pest dynamics but not much is known about these processes in dynamic smallholder production landscapes. This study investigated how seasonal land-use changes impacted *B. tabaci* and its parasitoids in an agricultural landscape in Uganda using a semi-manipulative field-experiment on smallholder farmers land. The dominant species of *B. tabaci* on cassava was Sub-Saharan Africa 1 (SSA1), which was also found on some other surrounding crops and weeds. The highest abundance of *B. tabaci* SSA1 nymphs in cassava fields occurred at times when landscapes had large areas of weeds, low to moderate areas of maize, and low areas of banana. These results are important to guide the development of land-use strategies that smallholder farmers can employ to manage whitefly pests.

**Abstract:**

Cassava (*Manihot esculenta* Crantz), an important commercial and food security crop in East and Central Africa, continues to be adversely affected by the whitefly *Bemisia tabaci*. In Uganda, changes in smallholder farming landscapes due to crop rotations can impact pest populations but how these changes affect pest outbreak risk is unknown. We investigated how seasonal changes in land-use have affected *B. tabaci* population dynamics and its parasitoids. We used a large-scale field experiment to standardize the focal field in terms of cassava age and cultivar, then measured how *Bemisia* populations responded to surrounding land-use change. *Bemisia tabaci* Sub-Saharan Africa 1 (SSA1) was identified using molecular diagnostics as the most prevalent species and the same species was also found on surrounding soybean, groundnut, and sesame crops. We found that an increase in the area of cassava in the 3–7-month age range in the landscape resulted in an increase in the abundance of the *B. tabaci* SSA1 on cassava. There was a negative relationship between the extent of non-crop vegetation in the landscape and parasitism of nymphs suggesting that these parasitoids do not rely on resources in the non-crop patches. The highest abundance of *B. tabaci* SSA1 nymphs in cassava fields occurred at times when landscapes had large areas of weeds, low to moderate areas of maize, and low areas of banana. Our results can guide the development of land-use strategies that smallholder farmers can employ to manage these pests.

## 1. Introduction

Invertebrate populations are influenced by a number of factors associated with both cultivated and uncultivated habitats in the landscape surrounding a target field [1]. In large-scale broad-acre cropping landscapes, farmers’ decisions around the choice of crop rotation heavily influence land-use patterns across multiple years [2]. In smallholder farming landscapes, however, land-use patterns often change due to highly flexible crop rotation plans that are dependent on factors such as family, village and regional demand for food and fiber. It is not currently clear how these dynamic land-use decisions impact pest and disease risk. Theoretically, agricultural landscapes may impact pest outbreak risk, firstly by providing extra host plant resources for pest populations (bottom-up effects) [3], and secondly by altering the ability of natural enemies to find and attack pest species (top-down effects) [4]. In farming landscapes where a species is polyphagous and has several crops and wild host plants suitable to support its population growth, the spatial and temporal arrangement of host plants may influence its abundance and outbreaks [5]. Additionally, peak densities of the species would be expected to be higher for low or non-suitable crops than for suitable crops as the suitable crops may have capacity to support high numbers of natural enemies [6].

Cassava (*Manihot esculenta* Crantz) is an important commercial and food security crop in many African countries where it is also ranked as a number one staple crop [7]. In smallholder production landscapes, cassava is planted as stem cuttings in fields approximately 0.1–1 acre in size. In Uganda, cassava is grown together with other crops such as banana, maize, groundnuts, beans, sweet potato, soybeans, and sesame either in the same field or in adjacent plots. Farmers may plant multiple cultivars of cassava and cassava fields can be of different ages that overlap in time owing to the multiple planting periods across the year. Insecticides are not easily accessible to most smallholder farmers. Thus, farmers rely on the use of manual cultural control methods and naturally occurring predators and parasitoids to manage pest outbreaks. Actively reducing the risk of pest outbreaks by manipulating the timing of planting and diversity of crops in the landscape is not commonly used, but represents a feasible option for farmers in these diverse farming systems.

The two major diseases of cassava, cassava mosaic disease (CMD) and cassava brown streak disease (CBSD), remain key factors limiting cassava production in Africa [8]. These diseases cause significant yield losses in Eastern and Central Africa [9], and are transmitted by the use of infected cuttings and vectored by whiteflies. The whiteflies in the *Bemisia tabaci* cryptic species complex (Hemiptera: Aleyrodidae) have been increasing in abundance on cassava over the past 20 years [10]. Management options to reduce *Bemisia* species densities in Sub-Saharan Africa are limited, but cassava varieties that are resistant or tolerant to the diseases have been developed [11,12]. Regardless, the increased densities of *Bemisia tabaci* cryptic species alone have been reported to reduce cassava yields by 40% [13]. Biological control by parasitoids and predators does occur but the impact of this is only beginning to be quantified [10,14,15,16] and cultural control options remained largely under-utilized. A recent study [17] that included cassava fields in Uganda, Tanzania and Malawi found that factors associated with the landscape surrounding the cassava field were unimportant for explaining variability in adult *Bemisia* density. However, the density of *Bemisia* nymphs and the parasitism of nymphs were heavily influenced by landscape factors, including the size of the cassava field, and total area of cassava in the landscape. This therefore suggests that by understanding these relationship(s), we may be able to identify landscape factors that could be altered by farmers to reduce the risk of whitefly outbreaks and cassava diseases.

In this study, we investigated how temporal changes in land-use across a season affect the abundance of *Bemisia* species and their parastioids in cassava fields in Uganda. Whilst Macfadyen et al. [17] examined the spatial patterns surrounding cassava fields across a broad geographic gradient (from Northern Uganda to Malawi), here we used a semi-manipulative field experiment to focus on the temporal patterns that characterize Eastern Uganda smallholder farms. With support from the farmers, cassava was planted in one-acre experimental plots in 10 landscapes. We chose to use this approach rather than relying on farmer-planted fields so we could standardize for cassava age, the cultivar used, and source clean cuttings, but still capture the diversity of production practices around the experimental fields that the farmers managed. We used a molecular approach to subsequently identify the *Bemisia* species on cassava and other plants in the landscape. We address the following research questions: (i) does the area of cassava and cassava plant age in the landscape correlate positively, at any point in time, with *Bemisia* abundance in the focal field; (ii) what combination(s) of the dominant crop types in the landscape correlate positively with *Bemisia* species abundance in the focal field; (iii) does an increase in the area of semi-natural habitats lead to increased parasitism of *Bemisia* nymphs in the focal field and therefore a decrease in *Bemisia* abundance? The term “focal field” is used here to refer to the cassava that was experimentally planted in the landscapes and therefore standardized for age, cultivar, and source parameters as different from farmer-planted cassava fields found in the surrounding landscapes.

## 2. Materials and Methods

### 2.1. Study Area

Landscape studies were undertaken in Kamuli district Figure 1 in south-eastern Uganda in the Kyoga plains agroecological zone [18]. This region is characterized by lowland rainfed conditions with an average annual rainfall of 1215–1328 mm. The altitude at the 10 landscape study sites varies from 1055 to 1109 m above sea level. The farming system prevalent in the district is rainfed annual cropping (usually two seasons per year) and predominantly with smallholder enterprises. The climate and seasonal environmental conditions are highly suitable for the growth and development of *B. tabaci* species, and there are often no environmental extremes within a season to limit population growth [19].

### 2.2. Planting of Focal Cassava Fields

With permission from the landholders, cassava was planted in one-acre trial plots in 10 different landscapes in Kamuli district, namely, Bugaga-Bulogo (1A), Buganza-Kasozi (1B), Butamwa-Buluya (1C), Mukokotokwa (1D), Tibwamulala-Butansi (1E), Bukafuga-Namwendwa (2A), Buganza-Namusagali (2B), Butamwa-Buluya (2C), Kasana-Mbulamuti (2D), and Tibwamulala-Butansi (2E) Figure 1. We could therefore standardize for cassava age, the cultivar used, and source clean cuttings (that have a low probability of infection with CMD or CBSD in the cutting material). The land surrounding the focal field was managed by the farmers as it would be under normal production practices. Virus-free cuttings of improved cassava varieties (Nam 130 and Nase 14) were planted in five fields at least 5 km from each other Figure 1 during the first season rains (April–May) and five during the second season rains (August–September). In total, 10 fields were planted with the two cassava varieties in equal quantities. Land-use information including crop, intercrop, growth stage and age of cassava plants (MAP = months after planting) was recorded at a minimum radius of 100 m from the centroid of the focal field. Land-use information was recorded at fortnightly intervals from one to four MAP, after which information was recorded monthly until harvest at 12 MAP. There was some temporal overlap at the end of season 1 and the start of season 2. Therefore, 1 MAP in season 2 coincides with ~9–10 MAP for season 1, hence they are the same point in real-time.

Field boundaries and landscape features were digitized and entered on Android tablets using offline maps and satellite image base layers, authored in ArcGIS Collector [20] and the resulting spatial layers were checked and cleaned in ArcGIS Desktop [21]. This information was confirmed by walking field boundaries and validating the details, which were then digitized and used to produce maps. We categorized 33 different land-uses and feature types including roads, buildings, crops, and non-crop areas. Land-use features from the maps were grouped into categories with similar features, namely, infrastructure, cropping, bare soil, livestock, native grassland, native shrubland, plantation and weeds. Polygons for the summarized land-use types for each site at each time point were generated. Figure 2 is a typical example of a landscape showing the various land-use features before planting (Time A11) and at one month after planting (Time A12).

### 2.3. Sampling Bemisia Species and Their Parasitoids

The sampling of *Bemisia* and their parasitoids occurred fortnightly from one to four MAP, then monthly until 12 MAP. Data were collected using the open data kit (ODK, University of Washington, Seattle, WA, USA) mobile data collection software (ODK Collect v1.4.12, Seattle, WA, USA) on predesigned electronic forms. Each plant was assigned a unique barcode that enabled easy cross-referencing of landscape, focal field, plant, and associated nymph and parasitoid data.

*Bemisia* adults, nymphs, and parasitoids on cassava were sampled on 30 randomly selected plants in each field at each time point (15 plants per variety). Adults were counted on the top-five fully expanded leaves per plant. For nymphs, an individual leaf (between the 8th and 10th positions) from lower down the stem with observable developed instars-at least 3rd instars, was collected, barcoded, packaged in a paper bag, placed in a cool box, and preserved for counting later in the laboratory. Nymphs were counted per leaf and categorized by instars (1st, 2nd, 3rd, and 4th) under a microscope. Parasitized and apparently healthy unparasitized nymphs were differentiated based on the position of mycetomes and the color changes in the nymphs. A parasitized nymph has displaced mycetomes while a healthy one has mycetomes directly opposite each other. Nymphs parasitized by *Eretmocerus* spp. have displaced mycetomes with shiny orange pupal skin (with pupae having red eyes), while those parasitized by *Encarsia* spp. have black pupal cases with brown meconia symmetrically located posteriorly on both sides. After counting individual nymphs under a microscope, the leaf was placed in an emergence container and incubated at room temperature for up to two weeks until the parasitoids or adult *Bemisia* emerged. The insects were placed in ethanol in a Petri dish and counted and identified under a microscope. Parasitoids were identified to genus level; namely *Eretmocerus* spp. and *Encarsia* spp. based on morphological characters [16]. Outside the cassava field, we conducted a timed search (15 min) on potential host crops in a 100 m radius from the centroid of the focal field and the presence or absence of *Bemisia* adults or nymphs on these crops was recorded. A decision was made on the area to conduct the search based on the availability of potential host plants as well as other refugia (thickets, herbaceous trees) in the 100 m radius from the centroid of the focal field.

### 2.4. Molecular Characterization of Bemisia Species Complex 

In the focal cassava fields, nymphs (on leaves) and adult *Bemisia* individuals were collected at 2, 4, 6, 8, 10 and 12 MAP for molecular identification. Nymph samples were also from several other crop types at 2, 6, 10 and 12 MAP, all preserved in 95–100% ethanol and stored for DNA extraction later. In the laboratory, nymphs were removed from leaves and separated into parasitized or unparasitized groups with the aid of a dissecting microscope. Unparasitized nymphs were counted into groups of 20 or 40 individuals representing individual populations and placed in individual sterile 1.5 mL Eppendorf tubes with 1 mL of >95% ethanol and stored at –18 °C until the DNA extraction stage. We used nymphs rather than adults for genomic (gDNA) extraction in order to link recovered nymphs to their plant hosts. Nymphs determined under microscope to be unparasitized may in actual fact have included some that were in early stages of parasitism (i.e., having a parasitoid egg or first instar parasitoid larva, all of which are not clearly detectable under the microscope). These individuals will have both whitefly and parasitoid DNA to enable *Bemisia* and parasitoid species identification by mitochondrial DNA cytochrome oxidase subunit I (mtCOI) sequence characterization.

#### 2.4.1. gDNA Extraction.

Nymphs were dried briefly (5–10 min) at 56 °C to remove all traces of ethanol and gDNA was extracted from nymph populations using a Qiagen Blood and Tissue DNA extraction (BTDE, Germantown, MD, USA) kit and followed the protocol as provided, and 4 µL RNase A (100 mg/mL; Qiagen) was added to remove RNA. Extracted gDNA was eluted twice in 100 µL (final volume 200 µL) of Elution Buffer (EB) (Qiagen). For downstream high throughput sequencing (HTS) of amplicon, we followed the protocol of Tay et al. [22].

#### 2.4.2. PCR and High Throughput Amplicon Sequencing

The concentration of individual gDNA samples was quantified and standardized to 5 ng/µL using EB buffer and kept at –18 °C until it was needed for PCR amplicon DNA library preparation for Illumina MiSeq NGS. The desired 3′ partial mitochondrial DNA cytochrome oxidase subunit I (mtCOI) gene widely used in molecular diagnostics of *Bemisia* species was amplified using specific HTS primer pairs [22]. Amplicon HTS runs were performed on Illumina MiSeq System, and sequence data processed using Trimmomatic [23] and analyzed as outlined in Tay et al. [22] using the Geneious software version 10.0.8. All mtCOI sequences were trimmed to 657 bp for molecular diagnostics of *Bemisia* species using reference sequences for Sub-Saharan Africa (SSA)1, SSA2, MED, IO and for pseudogene detection [24]. Detection of HTS-generated partial mCOI sequences of *Eretmocerus* and *Encarsia* parasitoids were identified using the reference sequences (GenBank MN646919, MN646949), prior to species identification against curated sequences in GenBank.

### 2.5. Data Analysis

The identity of land-use types present in each landscape (composition) was established while the amount (proportion) of each was calculated in Fragstats version 4.2 by taking the absolute area of each land-use type relative to the total area of the landscape [25]. We developed simple landscape metrics such as percentage cover of land-use types around the focal field (accounting for intercropping by establishing the percentage composition of intercropped patches and then calculating the area of each patch taking these intercrops into account), crop diversity, and the amount of non-crop vegetation (grassland or wooded areas). Crop diversity refers to the number of different crops in a 100 m radius from the centroid of the focal field. The area (m^2^) of each land-use type was calculated as well as the percentage cover of each land-use in the 100 m radius circle. To establish the effect of cassava age, we further classified cassava patches into 3–7 MAP, ideal for Bemisia growth and development or most suitable cassava; 0–3 MAP, young/less suitable cassava; greater than 7 MAP, old/least suitable cassava based on Sseruwagi et al. [26].

#### 2.5.1. Predictor and Explanatory Variables

We calculated response variables from our sampling data. For the research question (i) and (ii), we used the sum of *Bemisia nymphs* per field at each time point; for the research question (iii), we used the percentage parasitism of *Bemisia nymphs* per field at each time point. Percentage parasitism at the field-level was calculated as the sum of parasitized *Bemisia nymphs* divided by the total sum of *Bemisia nymphs* counted from that field, expressed as a percentage. The explanatory variables were age of cassava (MAP) and a site identifier, and the landscape factors relating to each of the crop and non-crop components of the landscape (Table 1). We checked for collinearity of the different combinations of explanatory variables using Spearman’s correlation coefficient prior to analysis. We used analysis of covariance (ANCOVA) to examine the significance of the interactions/correlations between *Bemisia* numbers and landscape factors (crop types used individually in models) using MAP as a covariate and using the “lm” function in the R “stats” package. For research question 1 (where the focus was cassava) and 3 (where the focus was the semi-natural areas in relation to parasitism), we used individual crop types in the models. For research question 2, we used two approaches; the ANCOVA that focused on individual crop types and the cluster analysis that focused on combinations of crop types (i.e., a multivariate approach). We also established (by Spearman’s rank correlation) the relationship between the different crop types and *Bemisia* nymph abundance in cassava. All the analyses were conducted in R 3.5.0 [27].

#### 2.5.2. Cluster Analysis

We used cluster analysis (i.e., a group of multivariate techniques to group cases of data (average nymph numbers) based on the similarity of responses to different landscape variables) as implemented in the “Class Discovery” data package in R. The clustering was done hierarchically (hclust function) using Ward’s method [28] that joined cases to clusters in a way that minimizes the within-cluster variance, with field and MAP as active input variables for the creation of clusters. The illustrative variables were crop types, non-crops and weeds, cassava age categories, while parasitized and unparasitized nymph counts per plant were the response variables.

Based on the minimum distance of the initial cluster, the 30 field time points in the landscape were grouped into seven clusters (see cluster dendrogram—Appendix A). We used analysis of covariance to investigate the effect of clusters, age of cassava and the various crop and non-crop landscape factors on the abundance of *Bemisia* and percent parasitism. A finer definition of cassava age was used in the cluster analysis. Four groups were identified: 1–2.5 MAP (“1early”), 3–5 MAP (“2peak”), 6–8 MAP (“3mid”), and 9–12 MAP (“4late”). We used the counts of *Bemisia* nymphs rather than adults (which move between plants) in analysis which were log (x + 1) transformed while counts of parasitized nymphs were log (x + 0.1) transformed prior to analysis.

## 3. Results

### 3.1. Identity of the Bemisia Cryptic Species

We analyzed 93 *Bemisia* nymph samples from cassava and two *Bemisia* samples from soybean using the HTS platform. We obtained 38,665 to 345,266 (75.3–98.9%) clean *Bemisia* sequences from the 93 cassava populations sequenced representing 2842 nymphs, with <3% unidentified sequences in the majority of populations. The dominant *Bemisia* species found on cassava in the different landscapes was the *B. tabaci* Sub-Saharan 1 species (SSA1, also called, SubSahAf1). The soybean *Bemisia* populations produced 148,810 and 164,843 clean reads with 97.7% and 98.2% assembled mtCO1 gene sequences matching to the SSA1 species, respectively.

Nymph samples from other host crops sequenced by the Sanger method showed a diversity of *Bemisia* species within the landscapes (Table 2). Various populations had sequences (between 2 and 24,403) with nucleotide identities (between 99.99 and 76.8%) matching to the hymenopteran parasitoid genus *Eretmocerus*, while low number (i.e., 0–100; <1%) of sequences showed identity to the *Encarsia* genus. Whilst *B. tabaci* SSA1 was dominant on cassava, it was also present on non-cassava host crops (soybean, groundnut, sesame). Other *B. tabaci* species (excluding SSA1) were found on sweet potato, pumpkin, sesame, beans, and *Euphorbia* weeds (Table 2).

### 3.2. Research Question 1: Does the Area and Age of Cassava in the Landscape at Any Point in Time Correlate Positively with Bemisia Species Abundance in the Focal Field? 

The proportion of cassava planted varied considerably between landscapes. In season one, cassava ranged from 14 to 33% of the landscape, and in season 2 from 7 to 46% of the landscape Figure 3A. The change in *B. tabaci* SSA1 nymph abundance with time in the different landscapes Figure 3B indicated generally higher numbers in season one than two, and a bimodal peak in season one between 2 and 4 MAP and between 7 and 10 MAP. In season two, a unimodal peak between 3 and 6 MAP was common across the landscapes.

While the extent (area) of cassava crops did not significantly correlate positively with the abundance of *B. tabaci* SSA1 in the landscape, the age of cassava plants was a significant factor that influenced the abundance of nymphs (Table 3). Young cassava was significantly negatively correlated to *B. tabaci* SSA1 abundance (F = 4.2, df = 1, 136; *p* = 0.04) while old cassava was marginally negatively correlated (F =19.1, df = 1, 136; *p* = 0.05). Only cassava in the ideal-age range (3–7 MAP) was positively correlated with the abundance of *B. tabaci* SSA1 (F = 3.82, df = 1, 136; *p* = <0.0002) (Table 3). Site 2B, however, had very high nymph numbers (mean = 472) despite a low proportion of ideal-age cassava in the landscape. The correlation between *B. tabaci* SSA1 abundance and cassava in the ideal-age range was positive (Appendix A) but we could see no obvious thresholds in terms of the area of cassava that leads to a dramatic increase in nymph abundance.

### 3.3. Research Question 2: What Combination of the Dominant Crop Types in the Landscape Correlates Positively with Bemisia Species Abundance in the Focal Field?

The common crops found in the different landscapes included cassava and several potential *Bemisia* host plants such as sweet potato, groundnuts, soybean, beans, and sesame. Besides coffee and banana which are perennial crops, the next most commonly planted crops in the landscapes were maize (0–39% in season 1; 0–61% in season 2), sweet potato (0–22% in season 1; 0–23% in season 2). The other planted crops after sweet potato were groundnuts, beans, soybean, and sesame, respectively. Other non-cassava crops planted in smaller areas included Bambara nuts, pumpkin, tomato, coffee, rice, millet, sorghum, cocoyam, and eggplant. Weeds (including a *Euphorbia* species) were present in the landscape. Percentage weed cover varied between landscapes from 0 to 41% in season 1, and from 0.8 to 56% in season 2. Across time, crop diversity varied between landscapes and between seasons. For example, diversity ranged between 4 and 14 crop types in season one landscapes, and between 2 and 10 crop types in season two landscapes. There was a higher diversity of crop types observed between 1 and 3 MAP, and 7 MAP in season 1, respectively, while this was true between 8 and 9 MAP in season 2 (Figure 4).

Different crops exhibited varied relationships with the abundance of SSA1 *Bemisia* nymphs in the focal field. These relationships were positive, negative, or non-significant (Table 3, Appendix A). For example, when the area of soybean, maize and beans increased in the landscape, the abundance of *B. tabaci* SSA1 in cassava reduced (Table 3, Appendix A). Conversely, an increase in the area of rice, eggplant, cocoyam, citrus, and banana in the landscape corresponded with an increase in *B. tabaci* SSA1 in the cassava field (Table 3, Appendix A).

A cluster analysis was conducted to group the landscapes according to the area of different crop types found. We identified seven clusters that broadly differed in the amount of maize and weeds surrounding the focal cassava field (Figure 5). The clusters showed a significant correlation to the abundance of *B. tabaci* SSA1 nymphs (as ordered in Figure 5). Whereas cluster seven had the lowest *B. tabaci* SSA1 abundance, cluster five had the highest abundance. The relative order of clusters (from lowest to highest) in terms of *B. tabaci* SSA1 abundance was 7, 6, 2, 3, 1, 4, 5, respectively.

The relative proportion of crop area between the clusters varied significantly, and this had a significant impact on the abundance of *B. tabaci* SSA1 in the landscape. Cluster 5 with the most nymphs (highest population) had a relatively higher proportion of weed cover at 54% with 22% under maize, 19% under banana, 14% under beans, 3% under soybean, and <1% under cocoyam. Cluster 4 with the second highest nymph population had 78% covered by weeds with 18% under maize, 3% under banana and 1% under beans. Comparatively, cluster 7 with the lowest nymphs comprised of 85% maize, the highest in all clusters, 7% soybean, 6% weeds and 2% beans (Figure 5). SSA1 *Bemisia* nymph population numbers differed significantly between clusters (F = 3.88, df = 6, 137; *p* = 0.001) and significantly by cassava age (MAP) (*p* = 0.007) with highest abundance in the 3–5 MAP group, followed by the 6–8 MAP (Figure 6).

Each cluster was made up of fields from a diversity of time points across the two seasons; however, there was some evidence of clustering by spatial location. The clusters differed in composition based on the spatial location of each field (site variable in Table 1). Clusters 2, 3, and 6 were composed of differing proportions of four of the 10 fields in the landscape studied. Clusters 4 and 5 had seven of the 10 fields represented while cluster 1 was the most diverse with eight of the 10 fields. Cluster 7 was the least diverse since it comprised a single field (2D) and is indicative of spatial variation.In general, cluster 1 had the highest number of observations (landscapes) followed by clusters 5, 2, 4, 7 and 6, respectively (Appendix A). The number of observations in a cluster arranged by MAP followed the same pattern (Appendix A).

### 3.4. Research Question 3: Does an Increased Area of Semi-Natural Habitats Lead to Increased Parasitism of Bemisia Species Nymphs in the Focal Field?

Non-crop areas which included semi-natural vegetation (grasslands or woodland) varied between landscapes and ranged between 0 and 60% in season 1 and between 3 and 52% in season 2 (Figure 7). The percentage parasitism of nymphs varied considerably across time in most landscapes (Figure 8). In season 1, the highest parasitism ranged between 26 and 42% late in the season (11–12 MAP). This percentage was highest in 1A fields and lowest in 1B fields. The lowest parasitism occurred in the earliest stages of cassava growth. Parasitism was much lower in season 2 than in season 1 and ranged between 26.9 and 34.5% with peaks between 5 and 8 MAP depending on the landscape (Figure 8). Parasitism was lowest in field 2B and highest in field 2C. The correlation between percent parasitism and the amount of non-crop area in the landscape showed a significant negative relationship (F = 4.34, df = 1, 142, *p* = 0.04). An increased area of non-crop, therefore, resulted in decreased parasitism of nymphs in the cassava field. Conversely, percent parasitism increased with an increase in the area of weeds, a relationship that was positively significant (F = 4.29, df = 1, 142, *p* = 0.04) (Appendix A).

## 4. Discussion

This is the first study quantifying the diversity of land-use changes that take place across a season in dynamic smallholder farming landscapes in Africa. By using an experimentally planted field, we could standardize the focal field for age and cultivar whilst allowing the surrounding landscape to change as it normally would under traditional smallholder production practices. We then linked the patterns in *B. tabaci* SSA1 nymph abundance to specific changes that take place within these landscapes. Our study has shown, firstly, that an increase in area of cassava fields in the ideal-age range results in an increase in the abundance of the *B. tabaci* SSA1 on cassava. Secondly, we found that the highest abundance *B. tabaci* SSA1 nymphs in cassava fields occurred at times when landscapes had the largest areas of weeds, low to moderate areas of maize, and low areas of banana. The mechanisms underlying these relationships are not clear and most likely involve processes such as colonization and oviposition behaviors of adults flying around these diverse landscapes [29], biotic and abiotic factors that cause nymph mortality, and management decisions made by farmers (although insecticide use is not common in these landscapes).

In smallholder production landscapes across Uganda, there was a year-round presence of cassava and other hosts plants that are potentially suitable for *Bemisia* species. From a comprehensive collection of nymph samples (93 cassava populations ~2842 nymphs), across fields and across time, and using a molecular method for identification, we can conclude that *B. tabaci* SSA1 is the predominant species in these landscapes. This result contrasts with cassava fields in Malawi that can have higher proportions of other species such as *Bemisia afer* and more diversity in terms of number of *Bemisia* species in cassava fields [17]. In our study, *B. tabaci* SSA1 was also recorded on other crops in the landscape, namely, sesame, soybean, and groundnut, which could be used as alternative host plants. Importantly, the molecular identification process we used was based on nymph samples rather than adults which can easily move between plants but not use them as a resource.

### 4.1. Research Question 1

Our analysis shows that if the patches of cassava in a landscape were not differentiated by their age, then there was no significant relationship to the abundance of *B. tabaci* SSA1 nymphs in the focal fields. However, when separated into three age-classes (young, old, and ideal), there was a positive relationship between nymph abundance in the cassava field and the area of ideal-age cassava in the landscape.

This relationship is likely to be the result of two processes: at field and at landscape levels. Firstly, at the focal field level: beside the increasing amount of ideal-age cassava plants in the landscape, the focal field itself also enters the ideal-age range, a suitable age which is highly attractive to *B. tabaci* SSA1. The increased amount of suitable-age cassava in the landscape leads to increased nymph numbers. The age of cassava plants plays an important role with respect to whitefly. Sseruwagi et al. [26] found that the population of *Bemisia* species peaked between 5 and 7 MAP and then dropped drastically as the plant became taller, woodier and less succulent. Several other changes in disease and virus characteristics in cassava have also been attributed to changes in the age of the crop (e.g., [30]. More recently, Kalyebi et al. [29] found that cassava fields of different ages supported different densities of *B. tabaci* SSA1.

Secondly, at the landscape level: as the amount of ideal-age cassava in the landscape increases, the total carrying capacity of the landscape increases, and this results in increased nymph numbers in all cassava fields. That is, there is an overall increase in the bottom-up resources (quantity and quality of host plants) for *B. tabaci* SSA1. Given the small size of our landscapes in this study, it is feasible that adults could move between the cassava patches in the landscape and select those most suitable for oviposition.

### 4.2. Research Question 2

When focusing on the composition and diversity of non-cassava crops, it is clear that there are types of landscapes that result in high *Bemisia* abundance. Those landscapes with the most *B. tabaci* SSA1 in cassava (cluster 4) had the highest proportion of weeds (78%), low–moderate maize area (18%) and low banana area (3%). In comparison, landscapes with the lowest *B. tabaci* SSA1 population in cassava (cluster 7) comprised of large maize areas (85%) and smaller weeds areas (6%). Cluster 7, however, consisted of only one field indicating low variation in its composition and highlighting the importance of spatial differences.

Species in the *B. tabaci* complex are polyphagous (e.g., [31] capable of using a wide range of hosts in the cassava landscape). In this study, *B. tabaci* SSA1 nymphs were recovered from soybean, groundnut and sesame demonstrating that the species can use alternative host plants for reproduction. Cluster 4 with the highest proportion of weeds, also had the highest population of *B. tabaci* SSA1 and indicates that weeds may be suitable hosts, and even a temporally limited increase in weed area could favor increased whitefly numbers in the cassava.

While other crops (i.e., eggplant, soybean [31,32] are known hosts for *Bemisia* species, the mechanisms by which they impact *Bemisia* species might be different. In our study, the amount of maize, for example, was negatively correlated with *B. tabaci* SSA1 abundance in cassava fields. To date, there has been no evidence of *B. tabaci* SSA1 species infesting maize in East Africa. Based on the plant apparency hypothesis [33], we hypothesize that maize may act as a barrier preventing *B. tabaci* SSA1 adults from accessing cassava host plants, thus decreasing the population of nymphs in cassava. A study from Ivory Coast found that intercropping maize with cassava reduced *Bemisia* species abundance [34], corroborating the results from our current study. However, Quintela et al. [35] reported the *B. tabaci* MEAM1 to be reproductively adapted to maize in Brazil following the expansion of the dry bean and soybean production. Based on this report of adaptation, *B. tabaci* SSA1 species may extend its host range in the future and could adapt to maize in Uganda. Comparing between *B. tabaci* MEAM 1 and *B. tabaci* SSA1 species is difficult as they are phylogenetically diverged (e.g., [36]). Furthermore, large areas of maize in the landscape may decrease the land available for suitable host plants, and therefore reduce the landscape-level host plant resources.

Many potential host plants in the landscape (see Table 2) showed a negative relationship with the abundance of *B. tabaci* SSA1 in cassava. Although some of these may be suitable host plants for *B. tabaci* SSA1, such crops may be relatively less suitable and less favored than cassava [31]. Field studies in diverse production landscapes are critical for understanding how this preference plays out. In a study in Colombia, cowpea intercropped with cassava significantly decreased numbers of whiteflies *Aleurotrachelus socialis* Bondar and *Trialeurodes variabilis* (Quaintance) [37] while an intercrop of mungbean and cassava reduced whiteflies significantly [38]. The actual mechanism by which this is achieved is not well understood although *B. tabaci* SSA1 is thought to be well-adapted to cassava [39]. For this reason, it may show a preference for cassava in the landscape and therefore it may be able to out-compete other whitefly species that also use cassava [31].

### 4.3. Research Question 3

An increased area of non-crop vegetation in the landscape did not lead to increased parasitism of *B. tabaci* SSA1 nymphs in cassava as has been observed in other farming landscapes [40,41]. Instead, *B. tabaci* SSA1 parasitism showed temporal and spatial variation across landscapes and seasons. Overall, parasitism rate was relatively high for an agricultural production landscape, reflecting the low use of pesticides on these farms. The relationship between parasitism and non-crop components of the landscape has been found to be inconsistent in other studies [42], although there are few studies in smallholder production contexts to refer to. The spatial study by Macfadyen et al. [17], which was focused on smallholder production landscapes, also found that several landscape factors influenced parasitism rate (not just area of non-crop vegetation).

The negative correlation observed between parasitism and area of non-crop in the landscape is an indication that parasitoid species may not be reliant on resources in the non-crop areas. This pattern does not preclude the use of resources outside the cassava field, but perhaps the cassava and other crops may be more of an assured source of food for the parasitoids than the semi-natural patches. In Asian cassava fields, within-field variables such as soil parameters and the plant phytopathogen status were found to be important for interactions between a cassava mealybug and its parasitoid [43]. Landscape factors did not affect mealybug abundance or parasitism rate [44]. In our study, the area of cassava was also negatively correlated to the non-crop area (correlation –0.24, Appendix A). Therefore, the parasitoids may be simply responding to *Bemisia* resources in the cassava fields.

In this study, we separated weed patches from semi-natural habitats, although they could both be considered non-crop patches. We found that weeds were positively and significantly correlated with parasitism rate (Appendix A) but were not significantly related to nymph abundance (Table 3). The diversity of weed plants may provide parasitoids with hosts, shelter, supplemental food resources, nesting locations, and/or sheltering sites both in space and time [40,45]. We also examined the net effect of parasitism rate, although we did not differentiate parasitoids. Nevertheless, there are potentially at least four species of parasitoids (i.e., *Eretmocerus mundus*, *Encarsia Sophia*, *E. mineoi*, and *Encarsia* sp-blackhead) that attack *Bemisia* species on cassava in Uganda [14], while our molecular diagnostics indicated likely high *Eretmocerus* and *Encarisa* species diversity in African cassava landscapes (also see Tay et al. [22]). There remains a need to better understand if different *Bemisia* species present in the landscape are attacked by different parasitoid species. This knowledge will be needed to enable these parasitoids to be developed as effective biocontrol agents.

## 5. Conclusions

Our results demonstrated that spatial and temporal changes in the surrounding landscape can impact both *Bemisia* species and their parasitoids in a focal field. This knowledge can be used to guide the development of cultural/landscape manipulation techniques that farmers can use to more effectively manage these pests. With infrequent insecticide use in these landscapes, habitat management techniques to promote or conserve parasitoids could provide options to better manage *Bemisia* populations. As established, whitefly population dynamics are governed not only by host plant availability and suitability but also by crop cultivation regime [46]. There is much variability in the cropping periods and sowing dates used by farmers with planting events spread over multiple time periods and planting seasons. Greater suppression of *Bemisia* species could be achieved by having greater cropping synchronicity across landscapes and establishing a fallow period [46]. Parry et al. [46] using a modeling approach found that overlapping plantings of cassava throughout a year resulted in the constant presence of young cassava in the landscape, which supported the highest populations. If farmers were able to manipulate cropping period practices and planting dates to alter the spatio-temporal distribution of host plants and the amount of cassava in the ideal age range in the landscape, a decrease in whitefly populations would be expected to follow. While current efforts to control *Bemisia* species are focused on the field scale, our study has highlighted the need to provide evidence for developing a landscape approach to manage this pest and the diseases it vectors.

## Figures and Tables

**Figure 1 insects-12-00269-f001:**
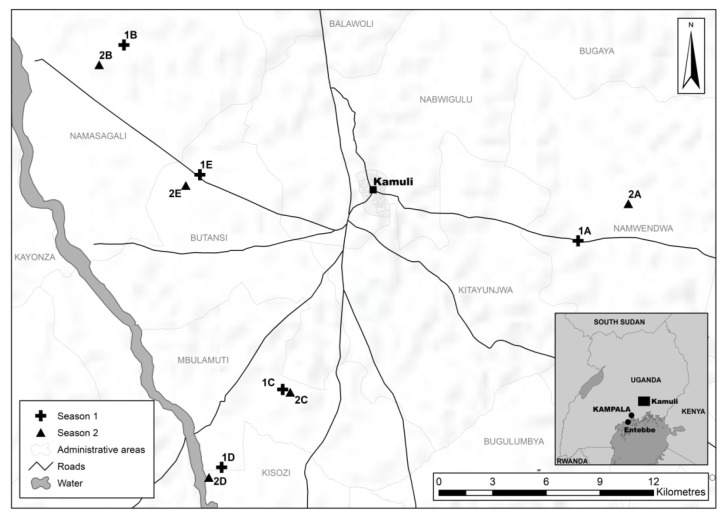
Location of the 10 experimental cassava fields (**1A**–**1E** and **2A**–**2E**) in different landscapes within Kamuli district–see inset, map of Uganda.

**Figure 2 insects-12-00269-f002:**
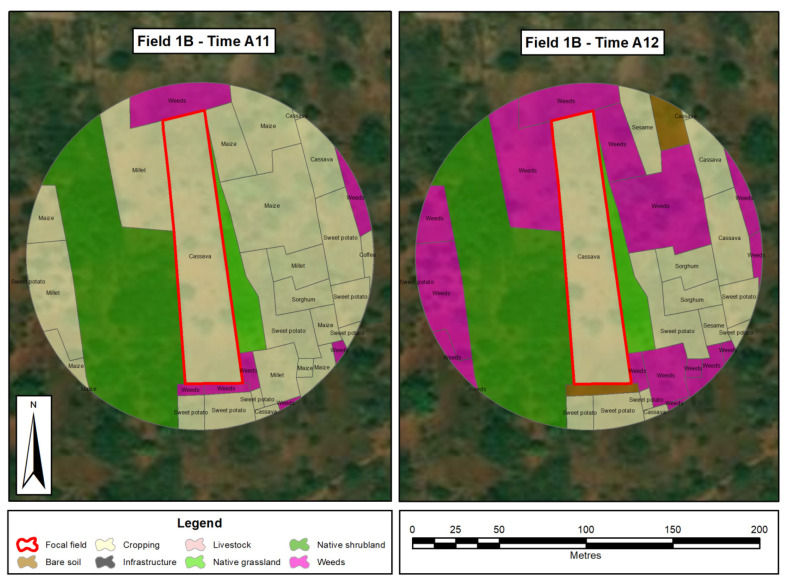
Land-use at a field site (1B) at time (Time **A11** and at Time **A12** one month later).

**Figure 3 insects-12-00269-f003:**
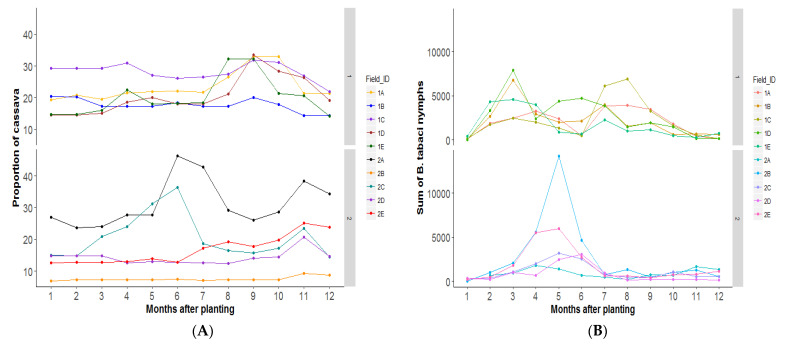
(**A**) Temporal changes in area planted with cassava in different landscapes in season 1 (March–February) and season 2 (August–July); (**B)** corresponding temporal changes in Sub-Saharan Africa 1 (SSA1) *Bemisia* nymph species abundance on cassava in the same landscapes in the two seasons.

**Figure 4 insects-12-00269-f004:**
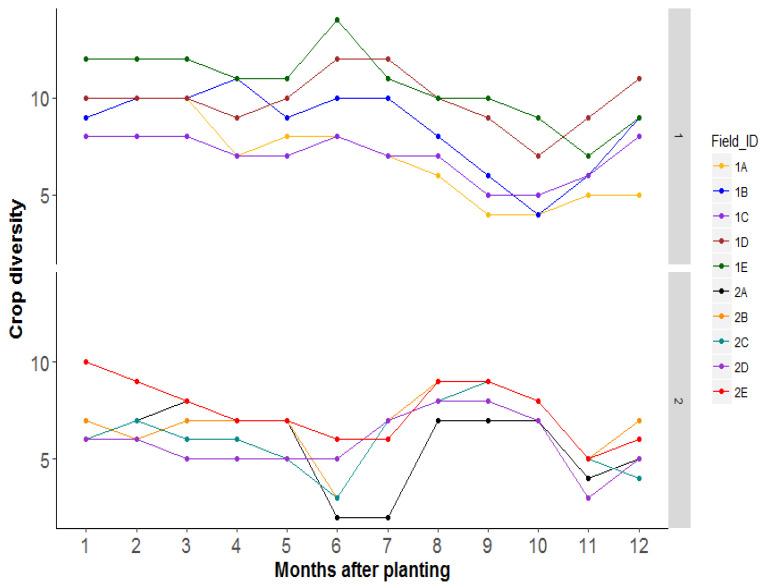
Changes in crop diversity in the different landscapes across time during season 1 (March–February) and season 2 (August–July).

**Figure 5 insects-12-00269-f005:**
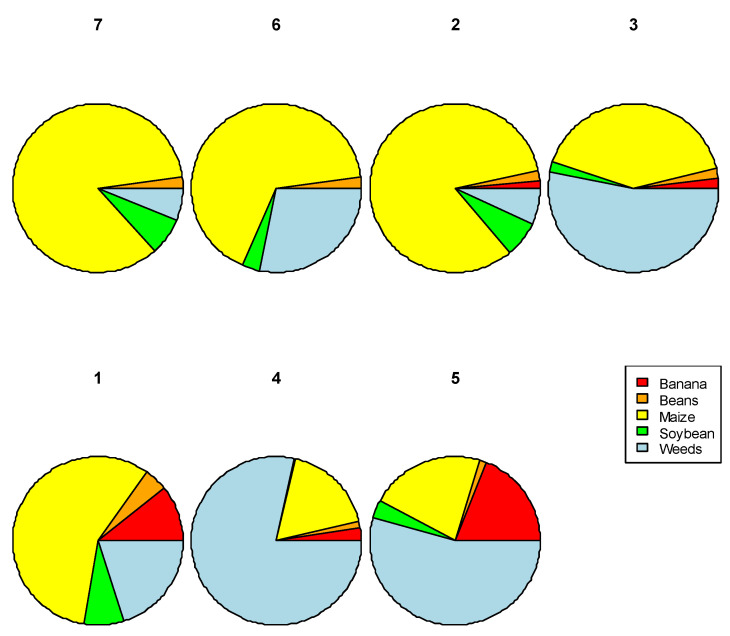
Clusters by crop type and planted area, relative to *B. tabaci* SSA1 abundance (from lowest to highest).

**Figure 6 insects-12-00269-f006:**
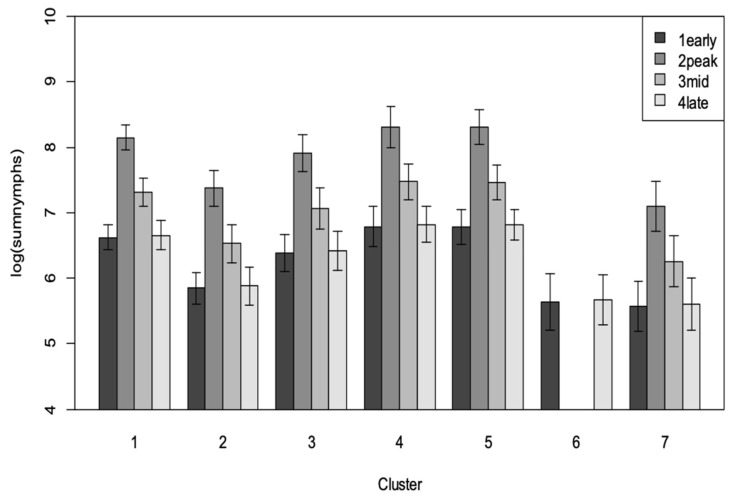
Abundance of *B. tabaci* SSA1 nymphs by cluster and cassava age (1early = 1–2.5 MAP, 2peak = 3–5 MAP, 3mid = 6–8 MAP, 4late = 9–12 MAP).

**Figure 7 insects-12-00269-f007:**
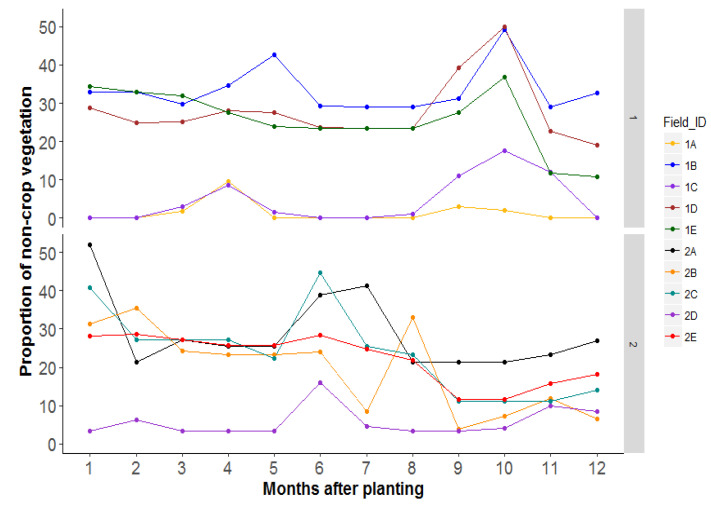
Change in non-crop (native vegetation) areas across time.

**Figure 8 insects-12-00269-f008:**
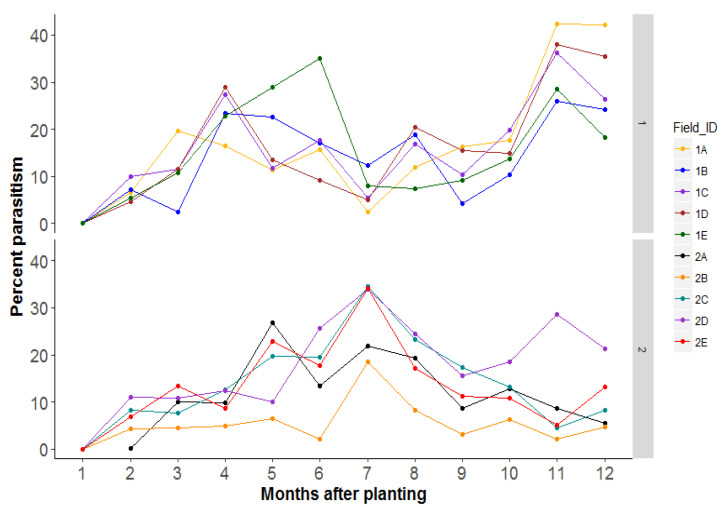
Change in parasitism level across time.

**Table 1 insects-12-00269-t001:** Predictor or explanatory variables and the research questions they were used to address in the data analysis.

Research Question	Name	Description
	**In-Field Factors**	
1,2,3	Time/Months after planting (MAP)	Represented as the age of the focal cassava (months after planting). There were 12 levels of this factor (1,2,3,4,5,6,7,8,9,10,11,12).
1,2,3	Field/site	These are the 10 different landscapes represented as (1A,1B,1C,1D,1E,2A,2B,2C,2D,2E) planted out during the first rains and second rains.
	**Landscape Factors that Change Across Time**	
1,2	Cassava	Percentage area covered by cassava ^†^
2	Sweet potato	Percentage area covered by sweet potato ^†^
2	Soybean	Percentage area covered by soybean ^†^
2	Groundnuts	Percentage area covered by groundnuts ^†^
2	Sesame	Percentage area covered by sesame ^†^
2	Beans	Percentage area covered by beans ^†^
2	Maize	Percentage area covered by maize ^†^
2	Coffee, Rice, Banana, Bambara, Pumpkin, Mango, Citrus	Percentage area covered by each of these crops ^†^
2,3	Weeds	Percentage area covered by weeds ^†^
3	Non-crop	Percentage area covered by non-crop (native vegetation/woodland) ^†^
1	Young cassava (less suitable)	Percentage area covered by cassava 0–3 MAP ^†^
1	Ideal cassava (most suitable)	Percentage area covered by cassava 3–7 MAP ^†^
1	Old cassava (least suitable)	Percentage area covered by cassava greater than 7 MAP ^†^

^†^ relative to total area within 100 m radius circle of the focal field.

**Table 2 insects-12-00269-t002:** Diversity of *Bemisia* species on other non-cassava crops in the study landscapes. The sample included nymphs only.

*Bemisia tabaci* Sample Code and Source Host Crop	Similarity with Published GenBank Sequences	Identity (%)
LF (Beans)	*Bemisia* Uganda1_KX570868	100
DF1 (Sesame)	Mediterranean_UG254_KX570768	100
CF ^†^ (Groundnut)	SubSahAf1_Uganda_KX570800	88.65
GF1 (Soybean)	SubSahAf1_Uganda_Masaka_AY903462	100
HF(*Euphorbia*)	SubSaharan Africa 13_KX570833	99.35
GF2 (Soybean)	*Bemisia* Uganda1_KX570858	99.85
IF (Sweet potato)	*Bemisia* Uganda1_KX570863	99.85
EF (Beans)	*Bemisia* Uganda1_KX570863	99.85
BF (Sweet potato)	Mediterranean_UG254_KX570768	99.70
MF (Pumpkin)	Mediterranean_Uganda_ASL_MH205754	100
KF (Sesame)	Mediterranean_UG254_KX570768	99.85
AR (Sweet potato)	*Bemisia* Uganda1_KX570868	100
NR (Soybean)	SubSahAf1_Uganda_KX570785	100
FR (Soybean)	SubSahAf1_Uganda_KX570785	100
DF2 (Sesame)	SubSahAf1_Uganda_KX570785	100

**Note:**^†^ indicates that significant amino acid residue differences exist between this partial mitochondrial DNA cytochrome oxidase subunit I (mtCOI) gene and the reported conserved amino acid residue patterns [24] indicating that this partial mtCOI sequence potentially represents a nuclear mitochondrial sequence widespread in *Bemisia* species.

**Table 3 insects-12-00269-t003:** Summary table of correlation coefficients (Spearman’s rank correlation) between land-use types in landscapes and *B. tabaci* SSA1 nymph abundance in cassava across time.

Crop Categorization	Relationship with Nymph Abundance in Cassava	*p*-Value
Cassava *	0.10	0.23
Cassava old *	–0.093	0.05
Cassava young *	–0.019	0.04
Cassava ideal *	0.36	0.0002
Maize	–0.40	<0.001
Beans *	–0.22	<0.01
Eggplant	0.28	0.002
Soybean *	–0.095	0.03
Sweet potato *	–0.22	0.0002
Rice	0.16	0.04
Banana	0.23	0.009
Citrus	0.17	0.02
Cocoyam	0.20	0.05
Bambara	–0.011	0.69
Coffee	0.76	0.035
Cowpea	–0.03	0.72
Groundnut *	0.019	0.02
Weeds	–0.049	0.56

* Plants that are considered host plants for the *B. tabaci* SSA1 and where nymphs were recorded during the study.

## Data Availability

The data presented in this study are available on request from the corresponding author.

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
