# Peer review of "Within-Season Changes in Land-Use Impact Pest Abundance in Smallholder African Cassava Production Systems"

_insects, 2021, doi:10.3390/insects12030269_

Round 1

Reviewer 1 Report

This was an interesting write-up of a study testing how whitefly infestations in cassava respond to the surrounding agricultural landscape over time. In general the manuscript is organized well. The focus on changes to the landscape within the growing season is interesting and under-studied, and I appreciate that the focal fields were standardized with experimental plantings to control local sources of variability.

The main issue to be resolved is the statistical analysis. In its current form the manuscript is ambiguous as to how exactly the models were structured and how the time-series nature of the data was being handled statistically. At some points I had a hard time following the logic from the statistics to the results which made it difficult to evaluate whether the conclusions were merited. See my comments with asterisks in particular.

Line by line comments. Some comments are just suggestions the authors can take or leave at their discretion.

105 ‘area size’ needs clarifying

Figures 2-3 The differences might be easier to see if these were side-by-side as two panels in the same figure.

*241 If the response variable was nymphs per field per time point, the samples are not independent because they are temporally autocorrelated. How is this accounted for with the statistical approach? (See next comment)

*250 Statistical methods are not clear. Like most readers I assume GLMM means ‘Generalized Linear Mixed Model’. If this is the case, be clear about which factors were included as fixed effects or random effects (was there a random effect for field ID, which would help account for the nested/autocorrelated design?). Also need to specify which type link function was used (binomial, poisson, etc.). Seems like for abundance a Linear Mixed Model (no link function) would be sufficient; for parasitism rate you might be able to use a Linear Mixed Model but could also use binomial GLMM if the response variable is expressed as the number parasitized vs. not parasitized instead of a rate. I also can’t tell which analyses were simply based on calculating a correlation coefficient vs. using regression. Maybe this could all be clarified if the statistical methods were separated out to match each of the three research questions.

*250 As written it is not clear to me how many predictor variables were included in each model. Table 1 lists a very large number of predictors. Were this many included in the GLMMs (In which case the ratio of predictor variables to observations seems very high)?  Or were they mostly used to lump the landscapes into groups with the cluster analysis?

*Table 1: not sure why months after planting was represented as a factor? It is continuous. Given that not all time intervals are equal (i.e. some are 0.5 months, some are 1), by making it a factor these differences are lost.

Fig 4-5 I think these figures might be more helpful if they focused on relationships between predictors of interest and your response variable(s) instead of just describing changes in cassava/Bemisia over time. If you choose to keep the figures as is (or relocating to the supplement), consider putting the two panels side by side instead of stacking. This will make differences along the y-axis more visually obvious. I also felt that color coding the individual fields and putting them in the legend was not necessary since the reader doesn’t know the difference between any of them anyway. It wouldn’t hurt to also include a summary line with mean and brackets for SE. You could still include the data from each individual plot but perhaps make them grey and superimpose a darker line with the mean.

*Table 3. Instead of saying the relationship is positive/negative/none, make it quantitative and specify the slope as well as other outputs from your model (e.g., standard error) Similarly, include all p-values even if not below your alpha of 0.05. Both 0.06 and 0.80 are ‘not’ significant but I interpret them quite differently.

Table 3 I also wondered if Tables 1 and 3 could be combined, especially if the descriptions of each variable were shortened.

Figure 7 / cluster analysis / supplement. It would be helpful to specify not only how many observations comprise each cluster, but how many landscapes (since there were several observations per landscape, a single cluster could be several landscapes or just the same one at several points in time). This is done to some extent in the paragraph beginning on 409 but I was still left wondering what proportion of each cluster was made up of repeat observations from the same landscapes. There might also be a clever way to combine Figs 7 and 8 (for example, put the pie charts along the x-axis of Fig 8).

427 Sentence conflates correlation and regression. Regression is the appropriate approach and what the methods suggest was done.

468 predominant

485 at the focal field level, do you mean that the focal field cassava is in the ideal stage simultaneously with cassava in the surrounding landscape? If so, it wasn’t clear to me as I read it.

496 I thought the sentence starting “That is…” is unnecessary, the previous one is clear enough

*548 This statement seems to be at odds with the results described on line 427.

Author Response

Reviewer #1

This was an interesting write-up of a study testing how whitefly infestations in cassava respond to the surrounding agricultural landscape over time. In general the manuscript is organized well. The focus on changes to the landscape within the growing season is interesting and under-studied, and I appreciate that the focal fields were standardized with experimental plantings to control local sources of variability.

Many thanks to the reviewer for these welcome comments.

105 ‘area size’ needs clarifying

Clarified

Figures 2-3 The differences might be easier to see if these were side-by-side as two panels in the same figure.

Side by side panels have been created in one figure (Figure 2), and figure 3 hence deleted. However, we do need assistance from the editorial team to make sure they are in the correct aspect-ratio. These are circles, not ovals.

*241 If the response variable was nymphs per field per time point, the samples are not independent because they are temporally autocorrelated. How is this accounted for with the statistical approach? (See next comment)

Analysis of covariance used in the analysis with months after planting (MAP) as a covariate accounts for the temporal autocorrelation between the samples.

*250 Statistical methods are not clear. Like most readers I assume GLMM means ‘Generalized Linear Mixed Model’. If this is the case, be clear about which factors were included as fixed effects or random effects (was there a random effect for field ID, which would help account for the nested/autocorrelated design?). Also need to specify which type link function was used (binomial, poisson, etc.). Seems like for abundance a Linear Mixed Model (no link function) would be sufficient; for parasitism rate you might be able to use a Linear Mixed Model but could also use binomial GLMM if the response variable is expressed as the number parasitized vs. not parasitized instead of a rate. I also can’t tell which analyses were simply based on calculating a correlation coefficient vs. using regression. Maybe this could all be clarified if the statistical methods were separated out to match each of the three research questions.

The matter of statistical approach has been clarified and we have been more specific in the text. The method used was ANCOVA (terminology corrected in the manuscript) which examines the significance of interactions/correlations between whitefly numbers and the crop types using MAP as a covariate. We used two statistical approaches to answer question 2. As we wanted to look at univariate responses of individual crop types, and then a multivariate approach that looked at clusters of combinations of crop types. For question 1, we were focussed just on cassava and for question 3 just on the semi-natural areas in relation to parasitism so we used the univariate responses. We have clarified this in the methods section by adding a column to Table 1.

*250 As written it is not clear to me how many predictor variables were included in each model. Table 1 lists a very large number of predictors. Were this many included in the GLMMs (In which case the ratio of predictor variables to observations seems very high)?  Or were they mostly used to lump the landscapes into groups with the cluster analysis?

We used ANCOVA to establish the significant correlations between crop types and whitefly numbers (simple univariate approach) and then we used a multivariate cluster analysis to group the data into classes of similar points based on a series of variables-landscapes and MAP just for research question 2.

*Table 1: not sure why months after planting was represented as a factor? It is continuous. Given that not all time intervals are equal (i.e. some are 0.5 months, some are 1), by making it a factor these differences are lost.

Data was collected at 15 time intervals (1, 1.5, 2, 2.5, 3, 3.5, 4, 5, 6, 7, 8, 9, 10, 11, 12) but we conducted  analyses at equal intervals (1, 2, 3, 4, 5, 6, 7, 8, 9, 10, 11, 12) with time (MAP) as a covariate. We have simplified this in Table 1 to just show what was analysed. Yes, we could have used MAP as a continuous variable, but chose to keep it as a factor in our analyses. Additionally, although the same plots were used every time plants were sampled, the samples were chosen randomly.

Fig 4-5 I think these figures might be more helpful if they focused on relationships between predictors of interest and your response variable(s) instead of just describing changes in cassava/Bemisia over time. If you choose to keep the figures as is (or relocating to the supplement), consider putting the two panels side by side instead of stacking. This will make differences along the y-axis more visually obvious. I also felt that color coding the individual fields and putting them in the legend was not necessary since the reader doesn’t know the difference between any of them anyway. It wouldn’t hurt to also include a summary line with mean and brackets for SE. You could still include the data from each individual plot but perhaps make them grey and superimpose a darker line with the mean.

We think the figures help to highlight the changes of cassava area within the landscapes as well as the changes of the cassava Bemisia nymphs, and so have been included as panels in one figure.  The corresponding figure on changes in mean nymph numbers reflects changes in total sum of Bemisia nymphs rather than the means and is clarified here as such.

*Table 3. Instead of saying the relationship is positive/negative/none, make it quantitative and specify the slope as well as other outputs from your model (e.g., standard error) Similarly, include all p-values even if not below your alpha of 0.05. Both 0.06 and 0.80 are ‘not’ significant but I interpret them quite differently.

Done.

Table 3 I also wondered if Tables 1 and 3 could be combined, especially if the descriptions of each variable were shortened.

We did try this, but it was hard to format clearly. To avoid clogging in the tables, the two tables have been left separate to maintain clarity in the descriptions.

Figure 7 / cluster analysis / supplement. It would be helpful to specify not only how many observations comprise each cluster, but how many landscapes (since there were several observations per landscape, a single cluster could be several landscapes or just the same one at several points in time). This is done to some extent in the paragraph beginning on 409 but I was still left wondering what proportion of each cluster was made up of repeat observations from the same landscapes. There might also be a clever way to combine Figs 7 and 8 (for example, put the pie charts along the x-axis of Fig 8).

Done, supplementary tables 1 and 2 have been added to this effect

427 Sentence conflates correlation and regression. Regression is the appropriate approach and what the methods suggest was done.

Corrected.

468 predominant

Corrected

485 at the focal field level, do you mean that the focal field cassava is in the ideal stage simultaneously with cassava in the surrounding landscape? If so, it wasn’t clear to me as I read it.

You are correct, there is a time when the two would be in the same ideal age range, and whitefly numbers would be expected to be higher than usual because of the suitability of the two. The sentence has been adjusted to make it clearer.

496 I thought the sentence starting “That is…” is unnecessary, the previous one is clear enough  

 “That is” has been deleted

*548 This statement seems to be at odds with the results described on line 427. We are unsure what is meant here but we are happy to correct if we can get further clarification.

Reviewer 2 Report

In the article entitled “Within-season changes in land-use impact pest abundance in smallholder African cassava production systems”, the authors investigated whiteflies and their parasitoids in cassava fields, focusing on landscape drivers of these insects and the role of temporal variations of the age of cassava plantings and the relative proportion of other crops and non-crop habitats. English writing is good and clear, and the article is concise, except for the results section that shows many figures and could be synthesized in a future version. I believe that the topic is very relevant for the management of this important crop for several African countries and the study can help to develop recommendations for farmers that could reduce pest populations. Nevertheless, I think that many aspects of the study, in particular related to the statistical analyses, need to be clarified and improved before the manuscript can be published. I would suggest major revisions and ask the authors to address the following suggestions.

Major comments:

-The introduction is clear and organized, but I wonder if a few more sentences could be added regarding what did you expect in terms of how other crops could influence whiteflies and parasitoids. Would you expect all crops to play a similar role as hosts of whiteflies, thus leading to a resource concentration in landscapes dominated by crops? Or is there evidence that some crops, in particular, would act as more relevant reservoirs of whiteflies? I would also like to read some more details about the community of natural enemies and why predators were not considered for your study. Was this choice only methodological or do parasitoids play a stronger role in controlling this pest?

-The number of fields is a bit low, especially considering that within each season only five fields were evaluated. However, I acknowledge that performing experimental sowing in these fields is laborious and that sampling within each field and across time provides a good database. On the other hand, was the 100 m scale chosen because of knowledge of the biology of the species? It seems like a small scale in comparison with most landscape-scale analyses.

That being said, I have several comments on the statistical analyses and the presentation of the results.

I agree with the use of GLMMs for your analyses. Where time/map and field/site used as random variables in these models (this is not entirely clear with the term “in-field factor”). If so, were they included as two separate variables, or was one nested within the other? Was cassava variety not included as a random or predictor variable? I imagined it could play a role but from what I understand it was not contemplated in the analyses?

These aspects about random factors and the results would be much clearer if more details about the models are reported. Table 3 only shows the significance of the predictors, but the estimates (and their SEs or CIs) are not shown, and adjusted relationships are also not shown for all cases. Figure S3, showing small correlation plots between variables is definitely not enough. I suggest including either a complete table showing these model parameters of a set of figures (or both) showing the predicted relationships plus raw data or partial residuals. In my opinion, this is more relevant than the figures showing variations in crop diversity or nymph abundance in every field, which could be included in the supplementary material. For example, the correlation between nymphs and cassava at different ages shown in Figure S2 does not show any clear trend, as opposed to the results of the models. Plotting the above-mentioned components of the models and data might help to show this better.

Another relevant issue is that no model selection was done, and the full model that was tested includes a large number of predictors. Even though there are many observations, these represent mainly pseudoreplicates of the ten fields and the models could be overparameterized. How was model fit evaluated? Even if correlations among variables were low, did you check if Variance Inflation Factors were low? Was spatial autocorrelation evaluated? As the design of the experiment seems to be paired across seasons, these could also be important.

I believe with so many predictors, a model selection would be advisable, by for example comparing models with different predictors (in the simplest way, by using the dredge function of the MuMIn package, starting from your full model) and finding the model/s that better explain your data.

On the other hand, these many predictors do not always show clear gradients, as seen in Fig. S3. The histograms suggest that some had relatively low variation among fields/time and might not be suitable predictors for these regression analyses.

-The reasons for using cluster analyses are not clear until the results are read. I believe that explaining why you use it in methods and perhaps also in the objectives can help to clarify this.

Minor comments:

-L 17: I think it may be better to say “the whitefly Bemicia tabaci” as you did in the abstract.

-L 30-31: it is not entirely clear what you mean by pest risk and the difference with pest populations, perhaps a small change of words can make this clearer.

-L 49-51: a reference to support this statement would be good, and none of the other references in the paragraph deals with crop rotation in particular.

-L 59-70: similar to my previous comment, I would expect at least one source for this information about the studied crop and the study region, if available.

-L 71: the two main diseases (at least in the region)? A quick search suggests that other diseases also attack this crop.

-L 96: perhaps the number of landscapes could be mentioned here?

-L 112-113: I would expect that the number of landscapes would be introduced here and in the next subsection you can describe the plots within each landscape.

-L 372-373: this sentence is not so clear, please rephrase.

-L 414-415: not entirely clear. If only one field was included in this cluster, isn’t the analyses really saying that this field was very different from the rest and had relatively low variation in its composition?

-L 572-574: but on lines 503-508 and 513-516 you stated that clusters with a high proportion of weeds had high nymph abundance and that weeds may be exploited by whiteflies. Please modify these parts for consistency.

Author Response

Reviewer #2

The introduction is clear and organized, but I wonder if a few more sentences could be added regarding what did you expect in terms of how other crops could influence whiteflies and parasitoids. Would you expect all crops to play a similar role as hosts of whiteflies, thus leading to a resource concentration in landscapes dominated by crops? Or is there evidence that some crops, in particular, would act as more relevant reservoirs of whiteflies? I would also like to read some more details about the community of natural enemies and why predators were not considered for your study. Was this choice only methodological or do parasitoids play a stronger role in controlling this pest?

More sentences and evidence to support the fact that some crops are more suitable for whitefly and natural enemies have been added. In this study, we focussed on parasitoids even though some predators are present in these systems. Research on predators was out of scope for our study, but it is important.

-The number of fields is a bit low, especially considering that within each season only five fields were evaluated. However, I acknowledge that performing experimental sowing in these fields is laborious and that sampling within each field and across time provides a good database. On the other hand, was the 100 m scale chosen because of knowledge of the biology of the species? It seems like a small scale in comparison with most landscape-scale analyses.

Our focus in this study was on temporal patterns, not spatial patterns between fields. Five fields, per season (10 in total) was enough given the frequency of samplings taken at the different time points and was sufficient to provide a temporal database that is robust and the most comprehensive in this context to date. The 100 m scale was chosen based on a related study (Macfadyen et al. 2020) that was a large-scale spatial survey in which we also used this same scale. This scale was found to be large enough to capture the diversity of crops surrounding the focal field, we were confident the whiteflies could physically move to all crop-types if they chose, and it was possible for us to map in a reasonable time frame. All the mapping was done manually, and as these are dynamic small-scale landscapes so we had to be practical about what we could achieve consistently across the seasons and across the large number of sites.

I agree with the use of GLMMs for your analyses. Where time/map and field/site used as random variables in these models (this is not entirely clear with the term “in-field factor”). If so, were they included as two separate variables, or was one nested within the other? Was cassava variety not included as a random or predictor variable? I imagined it could play a role but from what I understand it was not contemplated in the analyses?

Correct, we did not include cassava variety as a factor in analysis as it is being examined for another study. However, the varieties were consistent across the 10 sites and we wanted to focus on the temporal variation in the surrounding crop types in this study.

These aspects about random factors and the results would be much clearer if more details about the models are reported. Table 3 only shows the significance of the predictors, but the estimates (and their SEs or CIs) are not shown, and adjusted relationships are also not shown for all cases. Figure S3, showing small correlation plots between variables is definitely not enough. I suggest including either a complete table showing these model parameters of a set of figures (or both) showing the predicted relationships plus raw data or partial residuals. In my opinion, this is more relevant than the figures showing variations in crop diversity or nymph abundance in every field, which could be included in the supplementary material. For example, the correlation between nymphs and cassava at different ages shown in Figure S2 does not show any clear trend, as opposed to the results of the models. Plotting the above-mentioned components of the models and data might help to show this better.

The analysis has been clarified to the effect.

Another relevant issue is that no model selection was done, and the full model that was tested includes a large number of predictors. Even though there are many observations, these represent mainly pseudoreplicates of the ten fields and the models could be overparameterized. How was model fit evaluated? Even if correlations among variables were low, did you check if Variance Inflation Factors were low? Was spatial autocorrelation evaluated? As the design of the experiment seems to be paired across seasons, these could also be important.

We did check for spatial autocorrelation between some of the factors prior to modelling.

I believe with so many predictors, a model selection would be advisable, by for example comparing models with different predictors (in the simplest way, by using the dredge function of the MuMIn package, starting from your full model) and finding the model/s that better explain your data.

On the other hand, these many predictors do not always show clear gradients, as seen in Fig. S3. The histograms suggest that some had relatively low variation among fields/time and might not be suitable predictors for these regression analyses.

We agree that some factors had low variation and may not be the best predictors of whitefly abundance. We have tried to discuss these when we observed them such as the relationship between the amount of maize or weeds and Bemisia numbers. 

-The reasons for using cluster analyses are not clear until the results are read. I believe that explaining why you use it in methods and perhaps also in the objectives can help to clarify this.

Corrected. We have added details in the methods about why we used both a univariate and multivariate approach for Research question 2.

 Minor comments:

-L 17: I think it may be better to say “the whitefly Bemicia tabaci” as you did in the abstract. T

Corrected.

-L 30-31: it is not entirely clear what you mean by pest risk and the difference with pest populations, perhaps a small change of words can make this clearer.

This has been corrected to ‘pest outbreak risk’

-L 49-51: a reference to support this statement would be good, and none of the other references in the paragraph deals with crop rotation in particular.

References have been included that support crop rotation as contributing to changes in agricultural landscapes

-L 59-70: similar to my previous comment, I would expect at least one source for this information about the studied crop and the study region, if available.

A reference (FAOSTAT) has been included

-L 71: the two main diseases (at least in the region)? A quick search suggests that other diseases also attack this crop.

This has been changed to emphasize the “two diseases”

-L 96: perhaps the number of landscapes could be mentioned here?

The number of landscapes has been emphasized here as 10.

-L 112-113: I would expect that the number of landscapes would be introduced here and in the next subsection you can describe the plots within each landscape.  

Corrected.

-L 372-373: this sentence is not so clear, please rephrase.

Sentence rephrased

-L 414-415: not entirely clear. If only one field was included in this cluster, isn’t the analyses really saying that this field was very different from the rest and had relatively low variation in its composition?

This has been clarified and as inferred here, the field was very different suggestive of a spatial difference

-L 572-574: but on lines 503-508 and 513-516 you stated that clusters with a high proportion of weeds had high nymph abundance and that weeds may be exploited by whiteflies. Please modify these parts for consistency.

These sentences have been modified for consistency with the format and for coherence.

Round 2

Reviewer 1 Report

In general my comments have been addressed. Here are a few minor notes:

100 I think it is more appropriate to call this study ‘semi-experimental’ than a full-on manipulative experiment. The crop is standardized but was deployed into different extant landscapes that differ in composition. To be a true manipulative experiment the landscape would need to be changed intentionally for the purpose of the study.

Figure 2: If side by side maps are used, technically a legend and scale should only be required for one. Not a big deal; I’ll leave it up to the editor to decide.

269 the word ‘more’ is not needed

Table 3 Table would be more complete if it included test statistics, standard errors etc.

Author Response

In general my comments have been addressed. Here are a few minor notes:

100 I think it is more appropriate to call this study ‘semi-experimental’ than a full-on manipulative experiment. The crop is standardized but was deployed into different extant landscapes that differ in composition. To be a true manipulative experiment the landscape would need to be changed intentionally for the purpose of the study.

Corrected as suggested

Figure 2: If side by side maps are used, technically a legend and scale should only be required for one. Not a big deal; I’ll leave it up to the editor to decide.

Agreed. One legend and scale has been provided.

'269 the word ‘more’ is not needed

‘More’ has been deleted

Table 3 Table would be more complete if it included test statistics, standard errors etc.

The data in the table are correlation coefficients. This has been clarified in the table legend and not from anovas. As this is a non-parametric test, there are no standard errors, and only p-values are listed in the table.

Reviewer 2 Report

I thank the authors for their revised version of the article. I believe that some parts of the text were improved and their responses were also helpful to understand and justify some of their choices.

That being said, I still have some small suggestions and my major comments on statistical analyses were not considered. As reviewer #1 also shared many similar concerns, I encourage the authors to revise the quality of the analyses and improve the presentation of the results. I believe that this change would help to improve the manuscript considerably.

The new sentences in lines 262-270 are good to explain the analyses performed for each question, but several of the aspects reviewer #1 and I requested were not addressed. In the previous version of the text, the authors stated that GLMMs were used but now they changed it for ANCOVA. I believe this is a major drawback, as even using MAP as a covariate does not allow to consider the dependence among multiple samplings in a landscape. In my opinion, a GLMM with field ID and MAP as random factors would be the best choice and reviewer #1 stated the same. Table 1 actually indicates that these two were random factors, but that is not what the new description of the analyses suggests.

On the other hand, the results from question 2 (table 3) are extracted from separate analyses for each crop type? That is not completely clear on lines 266-268. While this might be better in terms of the potential overparameterization of including many different variables in one model, I think that some additional information could be obtained from these models. If the authors perform a model selection or even a model comparison by AIC values, it would be easy to determine which variables were more important and to establish relative preferred and non-preferred crops for whiteflies.

As pointed out by reviewer #1, Table 3 should also include standard errors or confidence intervals, and t/z values. Depending on their choice for modifying the type of analyses, other components would be relevant as well, such as the AIC values.

Also, in relation to the figures representing the results, I won’t insist on changing the figures of the main text if the authors prefer to show figure 4 as it is. But in the supplementary, I think that it would be better to show clear scatterplots of the relationships between crops and whiteflies (questions 1 and 2), including the fitted relationships from the models. Similar to Fig. S4 and S5, but for the different crops, perhaps limited only to those that were significantly associated with pest abundance.

Minor comments:

-L58-63: the newly added sentences are clear and improve the potential relationships between different crops and bottom-up/top-down control. However, as they are part of the first paragraph and B. tabaci is introduced later, the explicit example is not completely clear. Could this be mode more general and, if needed, detailed examples added below to keep a better order?

-Fig. 2: the side-by-side representation helps to better visualize the temporal variation. But as noted in your reply to reviewer #1, the aspect-ratio was modified and it should be broadened to keep the circular shapes and facilitate the reading of the labels.

-L 264: the function “lm” does not belong to the package “dae”, so please revise this detail when correcting the analyses.

Author Response

I thank the authors for their revised version of the article. I believe that some parts of the text were improved and their responses were also helpful to understand and justify some of their choices.

That being said, I still have some small suggestions and my major comments on statistical analyses were not considered. As reviewer #1 also shared many similar concerns, I encourage the authors to revise the quality of the analyses and improve the presentation of the results. I believe that this change would help to improve the manuscript considerably.

We acknowledge there is not one way to analyse this data set and we greatly appreciate the suggestions made by the editor and the reviewers. Please see our explanations as to why we have not changed the statistical analyses below. Please note we did engage with statisticians (Dr. Stephen Young, Dr. Marie Brueser) to help us with the analysis of this data set (after we struggled with other modelling approaches including GLMM) and this is the approach they suggested.

 The new sentences in lines 262-270 are good to explain the analyses performed for each question, but several of the aspects reviewer #1 and I requested were not addressed. In the previous version of the text, the authors stated that GLMMs were used but now they changed it for ANCOVA. I believe this is a major drawback, as even using MAP as a covariate does not allow to consider the dependence among multiple samplings in a landscape. In my opinion, a GLMM with field ID and MAP as random factors would be the best choice and reviewer #1 stated the same. Table 1 actually indicates that these two were random factors, but that is not what the new description of the analyses suggests.

 There are many ways you could potentially analyse this data set, and we appreciate the suggestions made by the reviewers. We are not trying to dismiss the importance of this issue; it is just challenging to address. We tried several modelling approaches prior to settling on ANCOVA and found this to be fit to our data set the best. We tried GLMMs and couldn’t get them to fit in a way that was useful before moving to an ANCOVA (the reason GLMM mistakenly appeared at first in methods). This may be a limitation of our skills in this area. However, we are unsure that a GLMM with MAP as a random factor will address the dependency issue identified (any more so than the way we have incorporated MAP in the ANCOVA). We have removed the statement about random factors from the table.

On the other hand, the results from question 2 (table 3) are extracted from separate analyses for each crop type? That is not completely clear on lines 266-268. While this might be better in terms of the potential overparameterization of including many different variables in one model, I think that some additional information could be obtained from these models. If the authors perform a model selection or even a model comparison by AIC values, it would be easy to determine which variables were more important and to establish relative preferred and non-preferred crops for whiteflies.

This is just not the approach we have taken in this data set and I’m not sure we could do it given the site number is equal to 10. Given the diversity of the landscapes we could easily end up with 6-7 landscape factors in each model (with MAP and site ID as random factors). We have used this model selection approach in another paper (Macfadyen et al. 2020) where we did include multiple landscape factors in the models. In that study we had large spatial variation across sites (as they came from a larger geographic range), and a larger n-value. It worked well for that study but may not be so useful here. The clustering approach we have used is a valid approach and we think is more useful for our data set, but we appreciate the reviewers suggestion and acknowledge that there is not one way to analyse this data set.

We have also clarified  in the analysis the use of  Spearmans rank correlation coefficient to measure the degree of relationship  between Bemisia nymph abundance and the different crop types  (shown in Table 3) in line 277-279.

 As pointed out by reviewer #1, Table 3 should also include standard errors or confidence intervals, and t/z values. Depending on their choice for modifying the type of analyses, other components would be relevant as well, such as the AIC values.

The data in the table are correlation coefficients (as clarified in the manuscript) and not from anovas. As this is a non-parametric test, there are no standard errors, and only p-values are listed in the table.

Also, in relation to the figures representing the results, I won’t insist on changing the figures of the main text if the authors prefer to show figure 4 as it is. But in the supplementary, I think that it would be better to show clear scatterplots of the relationships between crops and whiteflies (questions 1 and 2), including the fitted relationships from the models. Similar to Fig. S4 and S5, but for the different crops, perhaps limited only to those that were significantly associated with pest abundance.

Supplementary figures of scatterplots (S4-S9) showing the relationships have been included.

 Minor comments:

-L58-63: the newly added sentences are clear and improve the potential relationships between different crops and bottom-up/top-down control. However, as they are part of the first paragraph and B. tabaci is introduced later, the explicit example is not completely clear. Could this be mode more general and, if needed, detailed examples added below to keep a better order?

Corrected. Made more general, but the references still maintained

-Fig. 2: the side-by-side representation helps to better visualize the temporal variation. But as noted in your reply to reviewer #1, the aspect-ratio was modified and it should be broadened to keep the circular shapes and facilitate the reading of the labels.

Yes, agreed. We need help from the journal editors to correct this. -L 264: the function “lm” does not belong to the package “dae”, so please revise this detail when correcting the analyses.

Corrected

Round 3

Reviewer 2 Report

I thank the authors for their quick and detailed replies to my concerns.

While I believe that it is strange that GLMMs could not be fitted to their data, I acknowledge that there are multiple ways of exploring the data, and their explanations and description of the process are well explained.

My only recommendation is to carefully ensure that the quality and aspect ratio of the figures are ok for the final version, as at least in the pdf some look odd.